# Nightly Automobile Claims Prediction from Telematics-Derived Features: A Multilevel Approach

**Allen R. Williams [1], Yoolim Jin [2], Anthony Duer [2], Tuka Alhani [3] and Mohammad Ghassemi [1,\*]**

[1] Department of Computer Science & Engineering, Michigan State University, East Lansing, MI 48842, USA; will3670@msu.edu

[2] CSAA Insurance Group 3055 Oak Road, Walnut Creek, CA 94597, USA; victor.jin@csaa.com (Y.J.); anthony.duer@csaa.com (A.D.)

[3] Engineering Division, New York University Abu Dhabi, Saadiyat Campus, Abu Dhabi P.O. Box 129188, United Arab Emirates; tuka.alhanai@nyu.edu

\* Correspondence: ghassem3@msu.edu

**Abstract:** In recent years it has become possible to collect GPS data from drivers and to incorporate these data into automobile insurance pricing for the driver. These data are continuously collected and processed nightly into metadata consisting of mileage and time summaries of each discrete trip taken, and a set of behavioral scores describing attributes of the trip (e.g, driver fatigue or driver distraction), so we examine whether it can be used to identify periods of increased risk by successfully classifying trips that occur immediately before a trip in which there was an incident leading to a claim for that driver. Identification of periods of increased risk for a driver is valuable because it creates an opportunity for intervention and, potentially, avoidance of a claim. We examine metadata for each trip a driver takes and train a classifier to predict whether the following trip is one in which a claim occurs for that driver. By achieving an area under the receiver–operator characteristic above 0.6, we show that it is possible to predict claims in advance. Additionally, we compare the predictive power, as measured by the area under the receiver–operator characteristic of XGBoost classifiers trained to predict whether a driver will have a claim using exposure features such as driven miles, and those trained using behavioral features such as a computed speed score.

**Keywords:** telematics; usage based insurance; risk mitigation

## 1. Introduction

### 1.1. Motivation

Classical automobile insurance pricing is based on static features often indirectly and non-causally related to the risk propensity of the driver. There has been an increasing trend toward integrating driving habits and patterns into the insurance pricing equation. Policies which incorporate information about a policy holder's driving habits are often referred to as *usage based insurance* and can include policy types such as *pay-as-you-drive (PAYD)* which offers a lower premium pricing for drivers who drive fewer annual miles, and *pay-how-you-drive (PHYD)*, which uses features derived from a driver's actual driving patterns (e.g, braking event counts or cornering event counts) to assess a driver's claims propensity and determine a premium price (Arumugam and Bhargavi 2019). There are several advantages to usage based insurance including a fairer distribution of cost among the population of insured drivers, an expectation of savings on insurance costs for the majority of drivers (especially drivers in the lower income category), increased traffic safety through a reduction in miles driven, and an opportunity to provide feedback to drivers thereby reducing the total risk in an insurer's portfolio (Litman 2004; Tselentis et al. 2016).

Telematics data are continuously collected which adds a temporal dimension to the data in an insurer's portfolio. With this, it is now possible to not only improve the assessment of the claims propensity of a driver, but also to examine the relative risk a driver

poses in the near future. If a system could successfully determine whether a driver were at increased risk of an incident based on that driver's past driving patterns, an intervention could be made to hopefully nudge the driver back to their safer driving habits and prevent a potential incident from occurring. Data published by the insurance information institute show average losses of over $20,000 in 2020 for a bodily injury claim, with the costs of claims rising steadily year after year[1]. With such an increasingly large cost of payouts to claims, a system which could notify both a driver and the insurer that the driver may be at increased risk could be valuable to both parties in terms of money saved and injury avoided.

### 1.2. Objective

We were primarily interested in three research questions in this study. First, can we use telematic behavioral measures to determine which drivers were more likely to have a claim; that is, *can we assess the risk propensity of a driver based on the telematics derived features?* Second, can we determine which trips were more likely to result in claims *using only features derived from previous trips*? Finally, would the classification tasks be improved by combining the results of the driver risk assessment and journey risk assessment?

### 1.3. Literature Review

#### 1.3.1. Telematics Data

There are multiple ways of collecting telematics data for usage based insurance, and even more ways of processing these data into usable features for determining the risk propensity of a driver. Driven distance and location (e.g, miles driven in urban areas) have been used as measures of a driver's exposure to risk (Guillen et al. 2019). The exact relationship between distance driven and claims frequency has been debated, with some authors describing a learning effect, which offsets some of the increased risk exposure from additional miles driven (Boucher et al. 2017); however, it has been argued that the effect is due to a residual heterogeneity which is inappropriately captured by the model used, meaning that low distance drivers have certain traits that are different from high mileage drivers other than just their distances driven (Boucher and Turcotte 2020). The argument is that increased mileage should not be viewed as risk reducing, since a driver who decides to drive more miles has a fixed claims probability for their original driven distance and the additional distance can only increase that probability, although the marginal increase in probability of claims per mile may decrease with additional miles driven. Some studies have examined drowsiness detection using wearable headbands (Rohit et al. 2017), or fatigue using computer vision techniques (Abulkhair et al. 2015). There have been studies analyzing driver behavior using information acquired from a vehicle's CAN (controller area network) bus, detailing sequences of actions including braking and turning events, steering wheel angle, and vehicle speed, most often in an attempt to identify drivers based on the information obtained from the CAN bus. Carfora et al. used a combination of mobile GPS data along with information from the CAN bus to derive two clusters which they identify as highway driving and urban road driving (Carfora et al. 2019). So et al. (2021b) focused on generating synthetic telematics data which closely mimics the distribution it is trained on, while consisting of none of the same data points, an effort which they hope will make telematics data more accessible to researchers and accelerate the rate of progress in the analysis of telematics data.

Machine learning approaches have been applied to the task of predicting claims from features derived from telematics. Most papers have focused on the task of predicting which drivers are likely to have a claim. Pesantez-Narvaez et al. (2019) explored both logistic regression and XGBoost (eXtreme Gradient Boosting) on a small dataset to predict the existence of claims. They found that XGBoost performed favorably, but noted a difficulty in the interpretation of XGBoost when tree models are used as the base learner due to the difficulty in extracting model coefficients. They argue that the modest performance gain of XGBoost over logistic regression on their dataset did not justify its use over logistic regression given the relatively lower interpretability of results. They also note the need

for careful hyperparameter optimization with XGBoost due to its many hyperparameters and its ability to determine complex nonlinear decision boundaries. Several other authors have also had success using an XGBoost classifier for claims prediction using telematics data on datasets of up to 30,000 rows (Abdelhadi et al. 2020; Hanafy and Ming 2021). Alamir et al. (2021) use statistical preprocessing methods and compare support vector machines and random forest classifiers and find that random forest has the best prediction accuracy. Bahiraie et al. (2022) designed a classifier using tree genetic programming and found that they were able to outperform a kernel support vector machine on their dataset, using accuracy as their evaluation metric. So et al. (2021a) focused on designing a classifier using a version of AdaBoost that could handle the large levels of class imbalance in insurance datasets well and also provide insight into driving behavior. Our use of telematics data is different from the above in that we predict claims at multiple levels of resolution and recombine the individual classifier outputs in order to predict which journeys occur immediately before a claim, so that we can identify periods of relatively higher risk for a driver, in the hopes that an intervention can be made and the increased risk can be mitigated.

### 1.3.2. Imbalanced Data

Insurance claims prediction is an example of a classification task in which data are almost always strongly *class imbalanced*. A dataset is considered to be class imbalanced if one class occurs much more often than the others. In a binary classification task the minority class is often referred to as the positive class, while the majority class is called the negative class. The focus of this section and the remainder of the paper will be on binary classification rather than multiclass classification. Learning from imbalanced datasets presents several challenges and there have been a variety of tools developed to address these challenges. Many classifiers will learn the majority class more effectively than the minority class, and will misclassify examples of the minority class more often than they do examples of the majority class. Moreover in domains where imbalanced data are common, it is often the case that correctly classifying minority class (positive) examples is more important than correctly classifying majority class (negative) examples (López et al. 2013). Class imbalanced datasets often arise when the event that is to be detected is rare, so in addition to the relative scarcity of samples in the minority class, there may also be an inadequate number of samples of the minority class which creates a difficulty in distinguishing signatures of the minority class from dataset noise (Guo et al. 2017). See Figure 1 for an illustration of the class imbalance problem.

A related problem is class overlap. Overlap is said to occur when there are regions in the feature space with similar proportions of minority samples and majority samples. If a dataset is separable in the feature space then there is no class overlap; on the other hand, if examples of the two classes are distributed uniformly over the feature space then there is maximal class overlap. See Figure 2 for an illustration of the concept of class overlap.

Class overlap alone is a significant problem for classification however it presents even greater difficulties in the presence of class imbalance. Denil and Trappenberg (2010) explored the relationship between class imbalance and class overlap and found then when each was examined separately class overlap presented much more serious difficulties than class imbalance, and that the presence of *both* class imbalance and class overlap presented unique and catastrophic difficulties that did not occur with either class imbalance or class overlap alone.

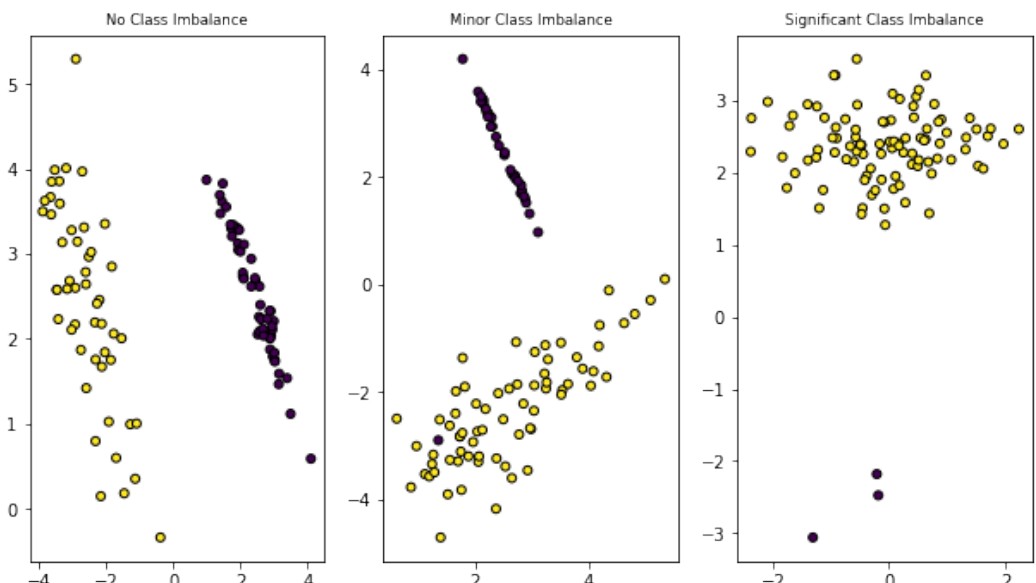

**Figure 1.** Plots of example datasets demonstrating varying levels of class imbalance. The different colors represent examples of different classes. In the leftmost plot the classes are completely balanced, the next two figures show progressively more severe class imbalance.

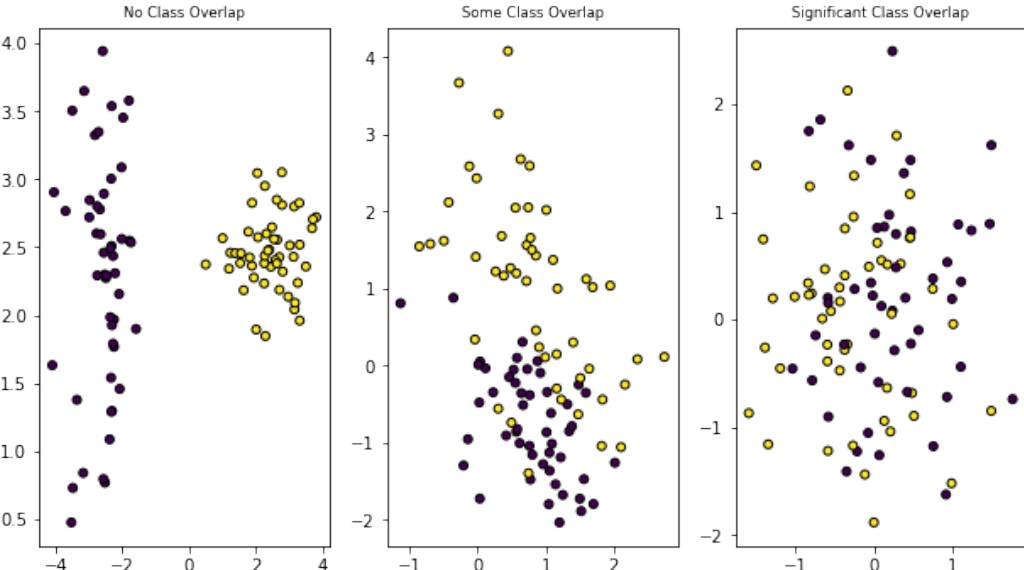

**Figure 2.** Plots of example datasets demonstrating varying levels of class overlap. The different colors represent examples of different classes. In the leftmost plot the classes are completely separable, in the center plot there are regions with similar densities of each class, creating difficult regions in the feature space for classification, and in the rightmost plot the classes overlap significantly, making them extremely difficult to successfully classify based on these features.

Class Imbalance and class overlap can be addressed at multiple different levels. At the data level, sampling methods can be used to increase the proportion of minority class examples. Some of the simplest sampling techniques are random over/under sampling, in which either some samples from the minority class are chosen at random to be repeated in the training set, or samples from the majority class are chosen at random to be omitted from the training set. A problem with the random oversampling approach is that it does not often aid in recognition of the minority class, rather forces the decision boundary to be focused on more specific areas within the feature space (Chawla et al. 2002). Alternatively

oversampling techniques such as SMOTE (Synthetic Minority Oversampling Technique) and its many variants produce new minority class samples thereby expanding the decision boundary and often aiding in the recognition of the minority class. SMOTE, in particular, works by selecting *k* nearest neighbors for each minority sample (where *k* is a hyperparameter) and for some proportion of these neighbors randomly choosing a point along the line connecting the minority sample with its neighbor and adding a new sample at that point, thereby augmenting the training data and expanding the decision boundary. See Figure 3 for an illustration of the SMOTE algorithm.

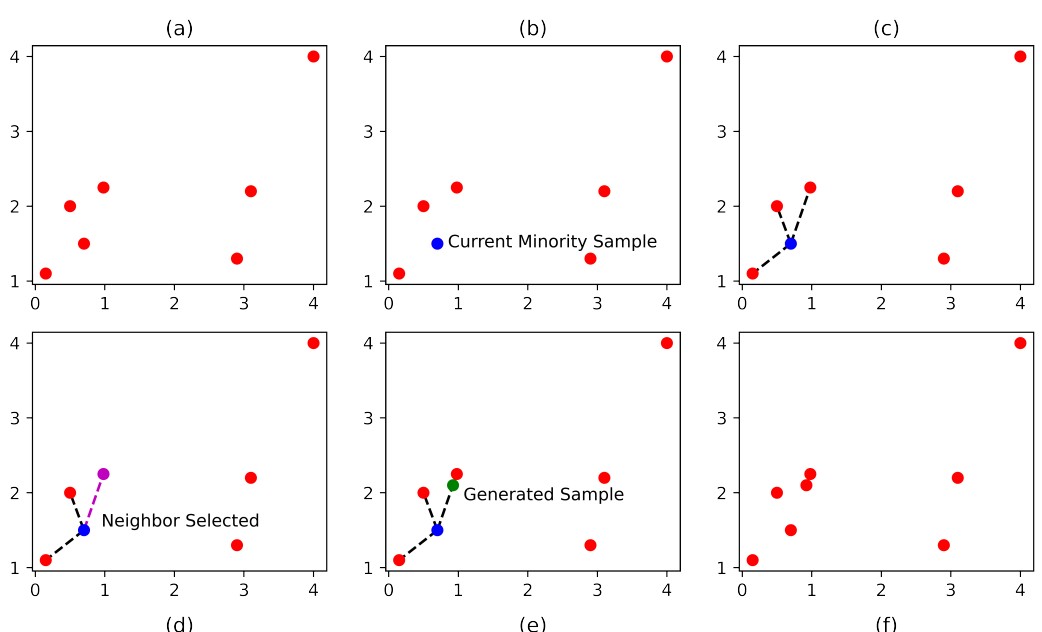

**Figure 3.** An illustration of the SMOTE algorithm process. Subplot (**a**) shows the initial population of samples in an area. These samples are all of the minority class. In (**b**) one of these samples is selected randomly as part of the algorithm (blue). In (**c**) the 3 nearest neighbors of this point are shown with the lines connecting the sample and its nearest neighbors. One of these neighbors is selected uniformly at random in (**d**), shown in purple. Next a point on the line is selected uniformly at random (**e**), shown in green, which becomes a synthetic minority sample. The new population is shown in (**f**).

Tomek link removal is a method for undersampling which can also help to address the class overlap problem (Kotsiantis et al. 2006). A Tomek-link is a pair of samples who are each other's 1−nearest neighbor but have opposite class labels. Formally, in the finite dataset $M \subseteq \mathcal{X} \times \mathcal{Y}$ where $\mathcal{X}$ is a population of samples and $\mathcal{Y} = \{0, 1\}$ the corresponding labels, write $n = |M|$, let $1 \leq i < j \leq n$, and let $M'_{i,j} = x_1, \ldots, x_{i-1}, x_{i+1}, \ldots, x_{j-1}, x_{j+1}, \ldots, x_n\}$, that is $M'$ is the projection of $M \setminus \{(x_i, y_i), (x_j, y_j)\}$ onto $\mathcal{X}$. Then, the pair $(x_i, y_i), (x_j, y_j)$ are a Tomek link if $y_i \neq y_j$ and

$$\forall x_k \in M'_{i,j} : d(x_k, x_i) > d(x_i, x_j) \text{ and } d(x_k, x_j) > d(x_i, x_j),$$

where $d$ is any metric, often Euclidean distance.

Another level at which the class imbalance problem can be addressed is at the classifier level. Often a weight is applied to samples in the minority class so that misclassification of minority samples is weighted more heavily in the loss function. Another commonly used method of dealing with class imbalance at the classifier level is ensemble classifiers. Most often these involve either bagging or boosting ensembles. Bagging (*Bootstrap Aggregating*) is an ensemble method in which a dataset is split into multiple datasets by sampling with

replacement (Bootstrapping) and models are trained on each of the bootstrap samples, then at prediction time the predictions of each of the models are aggregated in some way (this could be a majority vote, an average, or more complex aggregation methods). Boosting is another ensemble method that iteratively trains a collection of computationally efficient weak learners and aggregates their predictions to form a more powerful but still efficient ensemble. Boosting, along with weak learnability is defined formally in the PAC (*Probably Approximately Correct*) Learning framework (Valiant 1984).

A learning algorithm $A$ is said to be a $\gamma$-weak learner for the hypothesis class $\mathcal{H}$ if there exists a function $m_{\mathcal{H}} : (0,1) \to \mathbb{N}$ such that for every $\delta \in (0,1)$ and every distribution $\mathcal{D}$ over the feature space and every labeling function $f$ on the features, if the realizability assumption holds (i.e., there exists a hypothesis $h \in \mathcal{H}$ such that $\mathbb{P}_{x \sim \mathcal{D}}(h(x) \neq f(x)) = 0$), then if we have a sample set of $m > m_{\mathcal{H}}(\delta)$ examples drawn according to $\mathcal{D}$ then the output of $A$ will be a hypothesis $h$ whose true error will be, with probability greater than $1 - \delta$, less than $1/2 - \gamma$.

A learning algorithm $A$ is a strong learner for the hypothesis class $\mathcal{H}$ if for every $\varepsilon > 0$ there exists a function $m_{\mathcal{H}} : (0,1) \to \mathbb{N}$ such that for every $\delta \in (0,1)$ and every distribution $\mathcal{D}$ over the feature space and every labeling function $f$ on the features, if the realizability assumption holds, then if we have a sample set of $m > m_{\mathcal{H}}(\delta)$ examples drawn according to $\mathcal{D}$, then the output of $A$ will be a hypothesis $h$ whose true error will be, with a probability greater than $1 - \delta$, less than $\varepsilon$.

A boosting algorithm is one that can take a collection of weak learners and output a strong learner. A classic example of an efficient boosting algorithm is AdaBoost (Freund and Schapire 1997). AdaBoost iteratively trains *decision stumps*, which are decision trees of depth 1. A The weights of samples are chosen in each boosting round to incentivize the correct classification of samples which had been misclassified by the weak learners in earlier boosting rounds. The algorithm then assigns the weight of the tree in round $t$ to be $\frac{1}{2} \cdot log(\frac{1}{\varepsilon_t} - 1)$ where $\varepsilon_t$ is the error of the tree in round $t$. The final prediction then is a weighted sum of predictions of the trees from each of the rounds (Shalev-Shwartz and Ben-David 2014).

Another commonly used class of boosting algorithms, Gradient Boosting, of which XGBoost (eXtreme Gradient Boosting) is a particular example, trains each successive tree on the negative gradient of the loss function of the previous ensemble, then the output is a simple sum of the predictions of the tree in each boosting round, a concept known as *additive training* (Chen and Guestrin 2016). So if we denote the output of tree $j$ on input $x_i$ as $f_j(x_i)$, the first tree predicts the value of the label $y_i$ and the output of the ensemble after step 1 is $\hat{y}_i^{(1)} = f_1(x_i)$, the second tree will be fit to the negative gradient of the loss of tree 1, then added to the ensemble and the output of the ensemble will then be $\hat{y}_i^{(2)} = f_1(x_i) + f_2(x_i)$. The third tree will be fit to the gradient of the loss of the second ensemble and added to the ensemble itself. The process continues for $N$ rounds, where $N$ is a hyperparameter.

## 2. Methods

Herein we describe the data, methods, and validation approach used to develop models that assess the risk of driver- and journey-level claims using telematics data.

### 2.1. Data

All data for this study were sourced from CSAA Insurance Group. Our dataset consisted of 3,446,522 journeys taken by 5647 distinct drivers. For each journey, telematics features are derived from data collected from users' smartphones by the telematics data collection and processing company *The Floow*. The telematics data are represented as a collection of scores and event counts. We do not use any traditional insurance features such as age or gender in this study (Ma et al. 2018).

*2.2. Outcome*

For model development purposes, our outcome of interest was two binary variables that indicated the existence or absence of a claim at the driver and journey resolutions. A journey's class label is set to 1 if the following journey resulted in a claim for that driver while the driver's class label is set to 1 if the driver had a claim on any recorded journey. Our outcome was strongly class imbalanced; only 187 of the 3,354,578 journeys resulted in a claim, with 181 distinct drivers responsible for those claims. In our dataset there were 4 drivers with exactly 2 claims and 1 driver with exactly 3 claims. There were no drivers with more than 3 claims. We disregard any glass loss claims or claims where the driver is assigned less than 50% fault.

*2.3. Features*

The telematics data are represented as a collection of scores and event counts. The telematic scores summarize the driving habits based on characteristics such as a driver's tendency toward distraction, fatigue, and time of day—metrics which have been shown previously to be associated with risk propensity. Scores are computed at two different levels: globally (at the driver level) and locally (at the trip level). The trip level scores are computed for each trip taken at the conclusion of the trip and thus can change significantly even for the same driver on different trips, and the driver level scores are global scores for a driver summarizing his or her driving more succinctly. There are six scores for each driver, and six scores for each journey. The data providers of this study informed us that higher scores are considered better in all categories, i.e., a driver with an Overall score of 90 is considered to be less risky than a driver with an Overall score of 70, if all else is equal.

The telematic event counts are computed for each trip for categories such as braking, acceleration and cornering; for each of these events, an intensity level ranging from mild to extreme is also captured. We also use aggregate measurements such as the number of trips a driver has taken, or the number of miles the driver has driven as features for the assessment of driver risk propensity and the distance and duration of a trip for the assessment of the relative riskiness of the trip.

2.3.1. Driver Risk Assessment

To assess the risk profile of drivers we take into account the global scores provided for each driver, the number of miles driven, the number of discrete trips taken, and the days since the last heartbeat (a periodic signal used to verify connectivity between the driver's mobile device and the company's receiver). We divide these features into two distinct subsets, those derived from a driver's *behavior* and those measuring a driver's risk *exposure* (see Table 1 for a list of scores and summary statistics).

**Table 1.** Summary statistics for driver level scores and exposure measures in our dataset. Statistics are provided for the entire dataset, then separately for the group of drivers with at least one claim and the group of drivers with no claims.

| | Overall | Smooth Driving | Mobile Distraction | Time of Day | Fatigue | Speed | Driven Journeys | Kilometers Driven |
|---|---|---|---|---|---|---|---|---|
| **All Drivers (N = 5647)** | | | | | | | | |
| Mean | 80.61 | 71.30 | 86.33 | 75.53 | 83.59 | 79.90 | 418.15 | 6425.50 |
| Stdev | 5.28 | 8.54 | 9.14 | 3.41 | 9.17 | 5.18 | 435.79 | 8246.59 |
| Max | 100.00 | 100.00 | 100.00 | 93.28 | 100.00 | 100.00 | 5076 | 156,937.85 |
| Min | 28.89 | 12.07 | 36.38 | 38.81 | 42.86 | 51.13 | 1 | 0.80 |
| **Claims Group (N = 181)** | | | | | | | | |
| Mean | 79.58 | 69.61 | 85.03 | 75.09 | 84.84 | 78.72 | 658.17 | 9130.041 |
| Stdev | 4.74 | 7.45 | 9.18 | 3.38 | 7.19 | 4.93 | 506.12 | 8077.00 |
| Max | 92.83 | 86.45 | 100.00 | 91.27 | 97.81 | 92.83 | 2786 | 56,436.50 |
| Min | 61.05 | 40.35 | 52.73 | 61.05 | 63.04 | 63.70 | 1 | 4.18 |
| **No Claims Group (N = 5466)** | | | | | | | | |
| Mean | 80.64 | 71.36 | 86.38 | 75.55 | 83.54 | 79.94 | 410.21 | 6335.94 |
| Stdev | 5.29 | 8.56 | 9.13 | 3.41 | 9.23 | 5.18 | 431.04 | 8237.69 |
| Max | 100.00 | 100.00 | 100.00 | 93.28 | 100.00 | 100.00 | 5076 | 156,937.85 |
| Min | 28.89 | 12.07 | 36.38 | 38.81 | 42.86 | 51.13 | 1 | 0.80 |

### 2.3.2. Journey Risk Assessment

Journey scores are provided as a sequence of scores and event counts for each driver (see Table 2 for a list of behavioral scores and summary statistics).

**Table 2.** Summary statistics for journey level scores and exposure measures in our dataset. Statistics are provided for the entire dataset, then separately for the group of drivers with at least one claim and the group of drivers with no claims.

|  | Journey | Smooth Driving | Mobile Distraction | Time of Day | Fatigue | Speed | Duration (min) | Kilometers |
|---|---|---|---|---|---|---|---|---|
| **All Drivers (N = 3,446,522)** | | | | | | | | |
| Mean | 80.04 | 68.92 | 86.53 | 76.22 | 93.81 | 81.78 | 17.88 | 15.42 |
| Stdev | 9.19 | 12.85 | 14.99 | 6.72 | 6.66 | 10.05 | 25.00 | 30.89 |
| Max | 100.00 | 100.00 | 100.00 | 93.28 | 100.00 | 100.00 | 1876.6 | 1353.94 |
| Min | 13.90 | 6.98 | 21.88 | 21.85 | 42.86 | 37.50 | 0.0 | 0.0 |
| **Claims Group (N = 189,246)** | | | | | | | | |
| Mean | 79.65 | 68.34 | 86.05 | 75.86 | 94.21 | 81.71 | 17.03 | 13.81 |
| Stdev | 9.07 | 12.71 | 15.40 | 7.27 | 5.85 | 10.01 | 25.75 | 25.74 |
| Max | 100.00 | 100.00 | 100.00 | 93.28 | 100.00 | 100.00 | 1440.87 | 683.97 |
| Min | 17.27 | 6.98 | 21.88 | 21.85 | 42.86 | 37.50 | 0.43 | 0.00 |
| **No Claims Group (N = 3,257,276)** | | | | | | | | |
| Mean | 80.09 | 69.01 | 86.38 | 76.28 | 93.84 | 81.71 | 17.93 | 15.52 |
| Stdev | 9.12 | 12.72 | 9.13 | 6.66 | 6.61 | 10.07 | 24.95 | 31.165 |
| Max | 100.00 | 100.00 | 100.00 | 93.28 | 100.00 | 100.00 | 1876.60 | 1353.94 |
| Min | 13.90 | 6.98 | 36.37 | 21.85 | 42.86 | 37.50 | 0.05 | 0.00 |

Because we want to determine patterns in scores which are indicative of future claims we expand samples to include features from the previous 4 trips, including their labels, and we also provide 1st and 2nd order differences in scores to the classifier (see Figure 4 for an abbreviated illustration of the dataset widening process described).

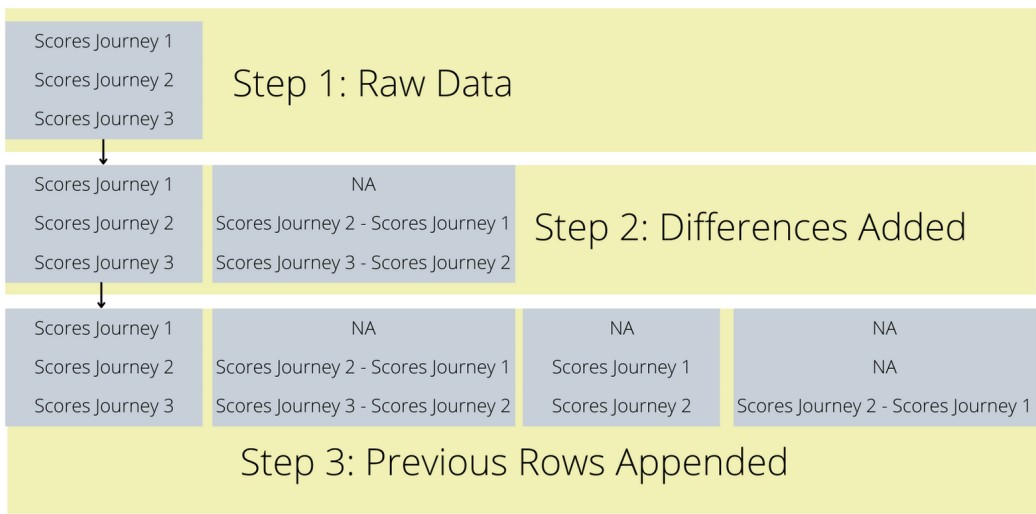

**Figure 4.** Illustration of the process used to create our dataframe. First differences are taken between features on trip $t$ and those on trip $t - 1$ (where t ranges from 1 to the total number of trips recorded for a driver, Note: The difference is not available if there are no data for trip $t - 1$ such as when $t = 1$), then the raw features and differences for previous rows is appended to each row. In our model differences were added again after step 2 for the the new difference columns created in step 2, and in step 3 the four preceding rows of features were appended to each row. This process multiplied the feature count by 15 and was followed by the feature selection process described in this section.

### 2.4. Model

We use XGBoost as the classifier to predict driver- and journey-level claims using the telematics features (Chen and Guestrin 2016). XGBoost was chosen for its simplicity of use, interpretability, and ability to natively handle missing data. The process of stacking rows and taking differences necessarily creates a lot of missing cells (see Figure 4), and in this setting the missing cells carry information. For example, the missing data created by taking differences and stacking rows conveys that there is no previous trip taken by the

same driver. For the driver classification task we train two XGBoost models on different features and stack their outputs with a final logistic regression classifier. We also examine the performance of weighted and l2-regularized logistic regression. For the regression model we apply a sample weight that is the inverse of the class distribution. Logistic regression models are fit using maximum likelihood, the problem is to solve

$$argmax_{\boldsymbol{\beta_0}, \boldsymbol{\beta} \in \mathbb{R}^d} \sum_{i=1}^{N} \left( y_i(\beta_0 + \boldsymbol{\beta}^T \boldsymbol{x_i}) - log(1 + e^{(\beta_0 + \boldsymbol{\beta}^T \boldsymbol{x_i})}) \right),$$

where $\beta$ are the model coefficients and $\beta_0$ is an intercept term. Often regularization is applied to the coefficients of the logistic regression, in our case we used *l*2 regularization which penalizes the *l*2 norm of the coefficient vector. The *l*2 norm of a vector, $\mathbf{w} \in \mathbb{R}^{\mathbf{n}}$, is

$$\|\mathbf{w}\| = \left( \sum_{\mathbf{i}=\mathbf{1}}^{\mathbf{n}} \mathbf{w}_{\mathbf{i}}^{\mathbf{2}} \right)^{\mathbf{1/2}},$$

so applying *l*2 regularization means instead of the maximum likelihood problem of unregularized logistic regression, we maximize the likelihood subject to the regularization term as follows:

$$argmax_{\boldsymbol{\beta_0}, \boldsymbol{\beta} \in \mathbb{R}^d} \left( \sum_{i=1}^{N} \left( y_i(\beta_0 + \boldsymbol{\beta}^T \boldsymbol{x_i}) - log(1 + e^{(\beta_0 + \boldsymbol{\beta}^T \boldsymbol{x_i})}) \right) - \lambda \|\boldsymbol{\beta}\| \right),$$

where $\lambda$ is a hyperparameter controlling the degree of penalization and is determined by cross-validation, and the intercept term is not penalized (Hastie et al. 2009). The optimal value of $\lambda$ can be estimated using the scikit-learn LogisticRegressionCV class as the estimator (Pedregosa et al. 2011). For the journey-level claims classification task we employ a model which combines the output of the driver and journey level classifiers. See Figure 5 for an illustration of our proposed final model for predicting journey risk.

Logistic regression comes with a natural means of extracting feature importances, namely its coefficients. By design logistic regression models the log-odds of an example belonging to the positive class as a linear function, that is, in logistic regression,

$$log \frac{\mathbb{P}(claims \mid X = x)}{\mathbb{P}(no - claims \mid X = x)} = \beta_0 + \boldsymbol{\beta}^T \cdot \boldsymbol{x},$$

where $\beta_0$ and $\beta$ are the intercept and coefficients estimated by the model respectively. Coefficients then give us the expected change in the *log-odds* of the outcome variable with respect to that coefficient. As our data were standardized prior to fitting the logistic regression model a hypothetical score with a coefficient of 0.31 could be interpreted as each standard deviation increase for this score is associated with an increase in the odds of a claim of $e^{0.31} = 1.36$, or 36%. Similarly, negative coefficients are associated with decreased odds of a claim (Hastie et al. 2009).

XGboost provides a variety of feature importance scores, including cover, weight, and gain. Gain refers to the improvement in classification results, as measured by XGBoosts's objective function, obtained by splitting a decision tree on a particular feature. In the case of XGBoost there are many decision trees, and the gain feature importance score for each feature is averaged across all splits in which that feature is used. See (Chen and Guestrin 2016) for an in-depth treatment of the objective function used by XGBoost and the types of feature importances it provides.

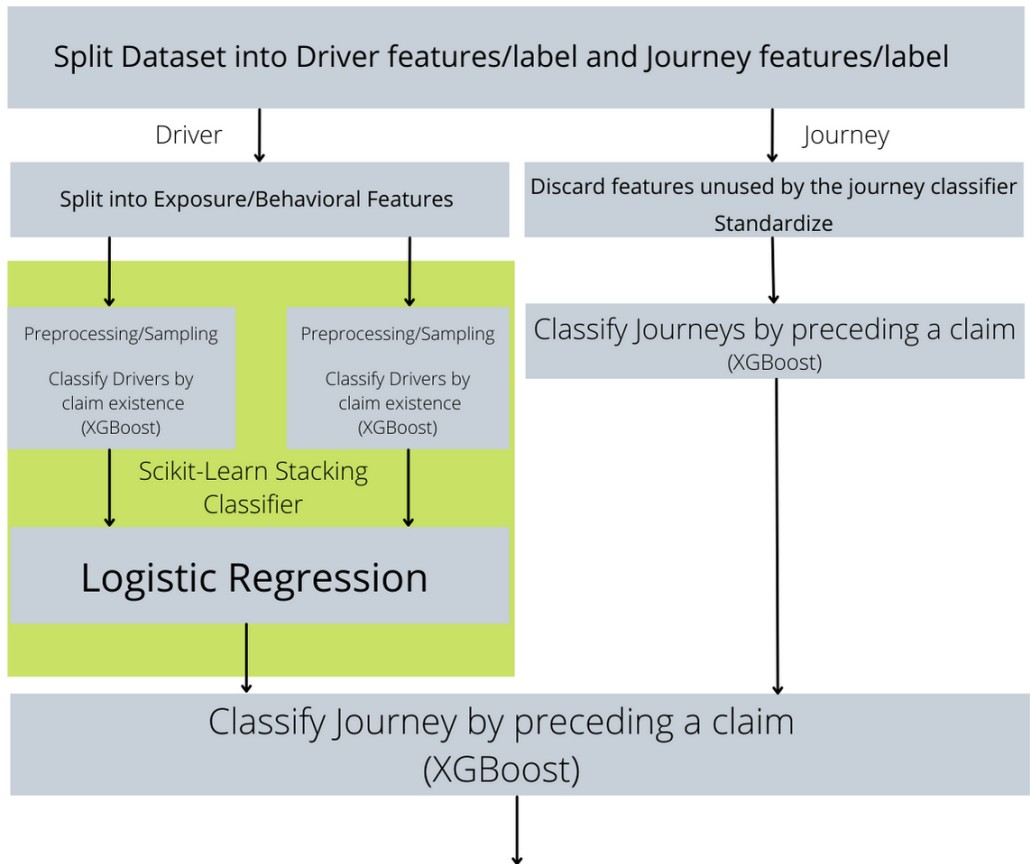

**Figure 5.** Illustration of the proposed model. The dataset is initially split into driver and journey feature sets creating two datasets of different sizes. The driver dataset is further decomposed into exposure features and behavioral features. XGBoost models are trained on each of these subsets of dataset and the two driver prediction models' outputs are stacked with a final logistic regression classifier. This stacking classifier is shaded in green. Finally the outputs of the journey risk classifier and stacked driver risk classifiers are stacked with a final XGBoost model which provides the final prediction.

### 2.5. Performance Metric and Validation Approach

The model is evaluated by area under the receiver–operator characteristic (AUROC), but cross-validation is performed across drivers rather trips, using *grouped* stratified 10-fold cross validation, where journeys are grouped by driver ID. Much like standard k-fold cross validation each journey is used exactly once in a test set, but in the *grouped* k-fold cross validation setting, a driver's trips are treated as a unit. Given an arbitrary driver ID, all trips taken by that driver ID will appear either in the test set or training set, and will never be split across the two sets. In this case it is also true that each *driver* appears in the test set for exactly one of the *k* folds.

Cross validation was performed using the scikit-learn StratifiedGroupKFold class. This method creates *k* empty folds (for *k*-fold cross validation) and greedily assigns each group to the fold which minimizes the standard deviation of class count across the folds after assignment. Although this method preserves the distribution of labels across the folds, it may produce unevenly sized folds. The result is a partition of the dataset into *k* test sets such that the class ratio in each test set is roughly the same, and roughly the same as that in the training set (Pedregosa et al. 2011). See Algorithm 1 for a pseudocode implementation of their algorithm.

---

**Algorithm 1** Group K-fold Cross Validation (scikit-learn).

---

**Input**: $\mathcal{X}, \mathcal{Y}, \mathcal{G}$ ▷ Feature set, associated labels, and vector of groups **Output**: A mapping of $i \in \{0, \ldots, k-1\}$ to the groups in the test set for fold $i$

**Require:** $\text{set}(\mathcal{G}) = \{0, 1, \ldots, m\}$
  **function** EVAL_FOLD(ys, fold)
     fold_y_counts[fold] += ys
     negs $\leftarrow [\,]$
     pos $\leftarrow [\,]$
     **for** $i \in \{0, \ldots, k-1\}$ **do**
        append fold_y_counts[$i$][0] / y_counts[0] to negs
        append fold_y_counts[$i$][1] / y_counts[1] to pos
     **end for**
     fold_y_counts[fold] -= ys
     **return** (std(negs) + std(pos))/2
  **end function**
  n $\leftarrow$ length($\mathcal{Y}$)
  group_y_counts $\leftarrow \{0 : [0,0], \ldots, m : [0,0]\}$
  y_counts $\leftarrow [0,0]$
  **for** $i \in \{0, \ldots, n\}$ **do**                    ▷ Count the labels in each group, and overall
     group_y_counts[$\mathcal{G}[i]$][$\mathcal{Y}[i]$] += 1
     y_counts[$\mathcal{Y}[i]$] += 1
  **end for**
  $\mathcal{G}' \leftarrow \text{set}(\mathcal{G})$
  sort($\mathcal{G}'$) in increasing order of standard deviation of label counts per group
  fold_y_counts $\leftarrow \{0 : [0,0], \ldots, k-1 : [0,0]\}$
  fold_groups $\leftarrow \{0 : [\,], \ldots, k-1 : [\,]\}$
  **for** $i \in \mathcal{G}'$ **do**
     best_fold $\leftarrow k+1$
     best_eval $\leftarrow \infty$
     **for** $j \in \{0, \ldots k-1\}$ **do**
        fold_eval $\leftarrow$ EVAL_FOLD(group_y_counts[$i$], $j$)
        **if** fold_eval < best_eval **then**
           best_eval $\leftarrow$ fold_eval
           best_fold $\leftarrow j$
        **end if**
     **end for**
     fold_y_counts[best_fold] += group_y_counts[$i$]
     append $i$ to fold_groups[best_fold]
  **end for**
  **return** fold_groups

---

### 2.5.1. Feature Selection

We reduce the dimensionality of the feature space by fitting the XGBoost classifier to the training data with a small subset of features chosen for each tree and selecting the features preferred by the classifier based on average gain of a feature across all splits. XGBoost determines a pool of candidates for the optimal split and chooses the best from among that pool of candidates, making the algorithm particularly robust to uninformative features (Chen and Guestrin 2016).

### 2.5.2. Addressing Class Imbalance

A significant challenge in this classification task is the class imbalance, as the vast majority of drivers have not had a claim. In order to address this challenge we use two sampling techniques: an oversampling technique called the Synthetic Minority Oversampling TEchnique (SMOTE), followed by undersampling technique called *Tomek link removal*. In our case, we remove the majority class member of each Tomek-link pair. We also make use of modified scikit-learn pipelines provided by the imbalanced-learn package, and many preprocessing utilities from the scikit-learn package (Kovács 2019; Lemaître et al. 2017; Pedregosa et al. 2011).

### 2.6. Combination of Journey-Driver Risk Assessment

After training the driver- and journey-level models separately, we trained a third model that combined the risk assessment from both models to provide an enhanced measure of risk at the journey-level.

That is, we use the driver risk scores estimated by the driver classifier, and the journey risk scores estimated by the journey classifier as input into a third classifier that predicts the same target as the journey classifier. We use an XGBoost classifier for this classification as well, and evaluate based on the same metrics with grouped stratified $10 - fold$ cross validation.

## 3. Results

Table 3 presents a snapshot of the results obtained by each classifier trained in this study. The following subsections provide individual results of the classifiers.

**Table 3.** List of Classifiers, their targets and AUROC achieved ($\pm 1$ standard deviation).

| Model | Target | Mean AUROC |
|---|---|---|
| Logistic Regression | Driver Claims | $0.69 \pm 0.10$ |
| XGBoost (Exposure features) | Driver Claims | $0.70 \pm 0.12$ |
| Stacking Classifier | Driver Claims | $0.70 \pm 0.11$ |
| XGBoost | Journey Risk | $0.59 \pm 0.03$ |
| Combined Classifier | Journey Risk | $0.62 \pm 0.05$ |

### 3.1. Driver Classification Results

Our final driver-level model had an AUROC of $0.70 \pm 0.11$ across the ten test-set folds, which we compare to the logistic regression score of $0.69 \pm 0.10$.

Additionally, we obtained an area under the receiver–operator characteristic in the driver classification task of $0.70 \pm 0.12$ using only exposure features (i.e, Driven Journeys, Driven Distance, and Heartbeat Days Elapsed). We were unable to achieve an AUROC higher than 0.6 using behavior features alone. The XGBoost model trained on exposure features performed slightly better than the logistic regression in terms of mean AUROC, and the stacking classifier had less variance in AUROC across the folds while performing on-par in terms of AUROC.

We find that scores are the most predictive in addition to either the number of driven journeys, or total distance driven in miles. See Figure 6 for an illustration of the relative feature importance in the driver classification models. Figures 7 and 8 show the receiver–operator characteristic curve for the XGBoost model trained on exposure features and the stacking classifier during cross validation.

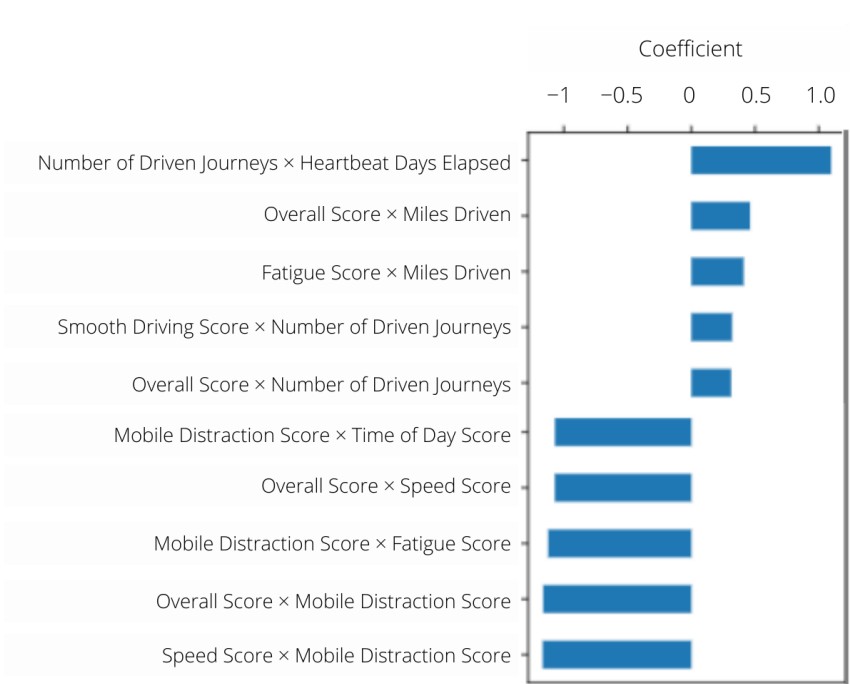

**Figure 6.** The five highest magnitude positive coefficients and the five highest magnitude negative coefficients for the penalized logistic regression model used in the driver prediction task.

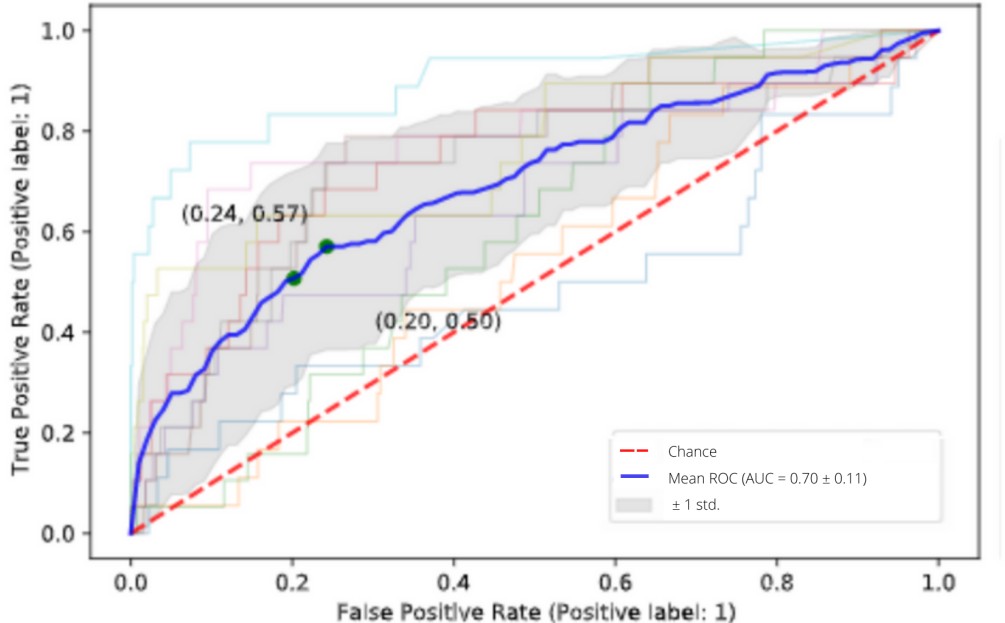

**Figure 7.** Receiver–operator characteristic over 10-fold cross validation for an XGBoost model trained only on exposure features in the driver classification task. The ROC for each of the ten folds is drawn in light colors. Mean AUC across the ten folds is 0.70 with the mean ROC curve drawn in dark blue, and $\pm 1$ standard deviation is shaded in grey.

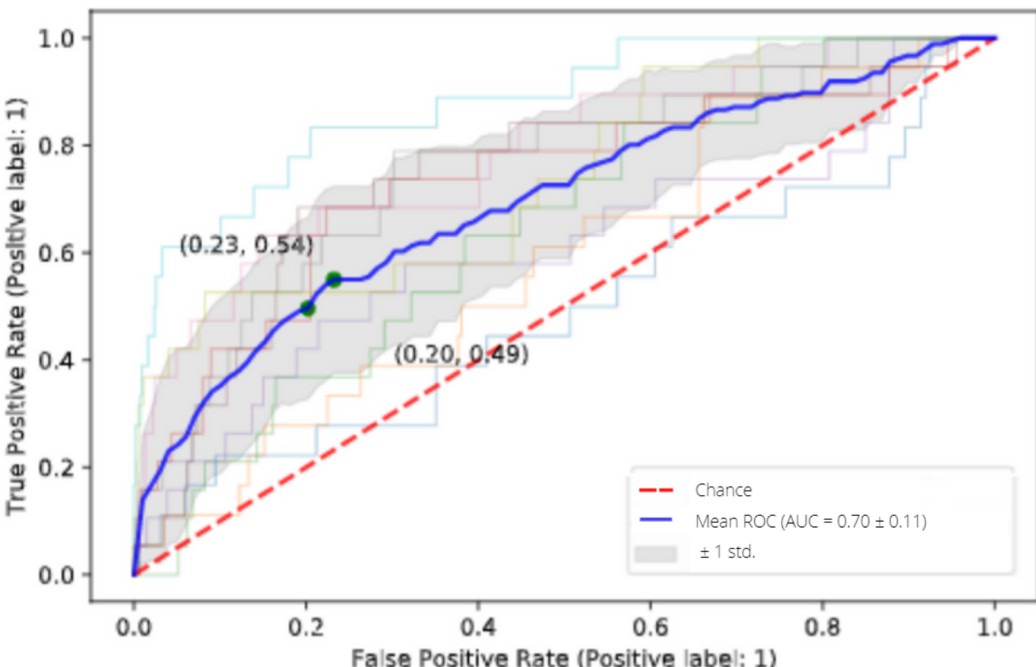

**Figure 8.** Receiver–operator characteristic over 10-fold cross validation for the stacking classifier in the driver classification task. The ROC for each of the ten folds is drawn in light colors. Mean AUC across the ten folds is 0.70 with the mean ROC curve drawn in dark blue, and ±1 standard deviation is shaded in grey.

*3.2. Journey Classification Results*

Our journey-level model had an AUROC of $0.59 \pm 0.03$ for the journey classification task in the absence of driver features. See Figure 9 for a listing of feature importance scores from an XGBoost model trained on journey features.

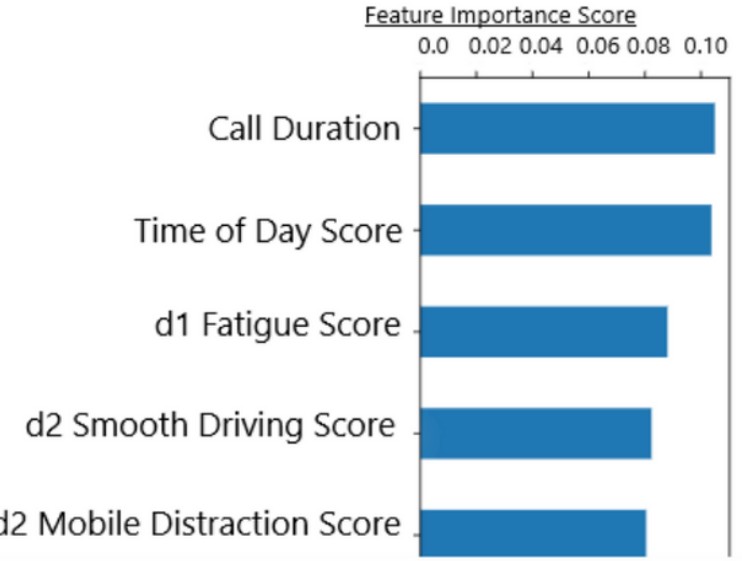

**Figure 9.** Top Five Feature importance scores for an XGBoost model trained on journey features. Here d1 denotes a first difference and d2 denotes a second difference, so the column "d1 fatigue score" consists of $NA, s_2 - s_1, s_3 - s_2, \ldots$ where $s_i$ is the score at time $i$, and the column "$d2$ Smooth Driving Score" consists of $NA, NA, s_3 - 2s_2 + s_1, s_4 - 2s_3 + s_2 \ldots$ where again $s_i$ is the score at time $i$. Some values are $NA$ since the difference is undefined if either of its terms are undefined.

Combination of Journey-Driver Risk Assessment Results

We report an area under the receiver–operator characteristic of $0.62 \pm 0.05$ with the majority of the importance placed on the journey risk rather than the driver risk. See Table 3 for a summary of classifiers used, their targets, and area under the receiver–operator characteristic achieved.

## 4. Discussion

In this paper we have presented a model which is able to predict the likelihood on the next trip a driver takes with an AUROC of 0.62. We have also explored a stacked model composed of XGBoost classifiers which was able to slightly improve the AUROC and tighten the ROC variance compared to a classifier trained on exposure features alone. We have shown that telematics derived features are predictive of claims propensity for drivers, and that the risk of a claim can be predicted in advance with some level of accuracy, even with data computed only nightly rather than real-time data. Ultimately the prediction of claims risk in the near-future would allow an intervention and ideally a prevention of the claim. We note the significance of the work, some limitations and exciting possibilities for future work below.

### 4.1. Significance

Much of the work in telematics has been around the derivation of features from raw data or the use of features to classify risky drivers for insurance premium pricing. We have shown that these features can be used to predict claims in advance, which opens the possibility of preventing that claim. Claims range in severity but can cost tens of thousands of dollars and can result in the injury or death of the driver or others. Since telematics data are already being collected and stored such a claims prevention model would be inexpensive to implement but could have a great benefit for both the insurer and the insured.

### 4.2. Limitations

The most important limitation of the present work is that it used black-box scores as features for classification, and data provided only at the trip level, rather than in real-time. Although our model was able to predict risky trips with a surprisingly high AUROC using trip level data, the examination of real-time driving behavior would allow the exploration of patterns in driving behavior immediately before incidents resulting in claims occurred and could further improve the classificaton results. Secondly, our risk analysis is based on population level statistics, which could be improved by using a more individual and personalized assessment of risk. There is an analogy with personalized nutrition, where it is likely that personalized advice may outperform advice based solely on population level statistics (Jinnette et al. 2021). In our case this introduces the questions:

- *does influencing a driver away from patterns associated with claims at the population level decrease the claims propensity of that driver?*
- *how can we best combine insights from population level studies and and personalized advice derived from a drivers own data only?*

### 4.3. Future Directions

An interesting direction for future research would be to experimentally test whether influencing drivers toward scores associated with lower risk could truly reduce their claims' propensity. Experimentally testing intervention strategies could address questions such as:

1. After identifying that a driver is at increased risk, what is the best method of intervention?
2. Does the timing of the intervention influence its effectiveness in reducing a driver's risk?
3. Can we identify the reason why the driver was considered to be at risk, and personalize the intervention to that individual? (e.g, are there signatures in the data indicative of anger, alcohol consumption or extreme stress?)

Another exciting direction for future work is the expansion of the statistics used in the classification of the journey risk. In our case, we used differences and stacked frames to allow the classifier which expects IID (Independent and Identically Distributed) tabular data to use the sequences of data.

Finally we found that oversampling was helpful in classifying driver risk, especially when using behavioral variables, where the minority class was oversampled to produce approximately a 1:3 ratio of positive to negative samples, but due to the grouped and sequential nature of the journey classification task, we did not use any oversampling for the journey classification task. A suitable combination of oversampling and undersampling techniques may be of great use on such an imbalanced dataset. It would be worthwhile to explore dataset augmentation with generative sequence models that fit this use case and examine their efficacy as a preprocessing tool in the journey prediction task.

**Author Contributions:** Conceptualization, A.R.W. and M.G.; methodology, A.R.W., M.G. and T.A.; software, A.R.W. and Y.J.; validation, A.R.W. and M.G.; investigation, M.G. and A.R.W.; resources, Y.J.; data curation, A.D. and Y.J.; writing—original draft preparation, A.R.W.; writing—review and editing, M.G. and A.D.; visualization, A.R.W.; supervision, M.G.; funding acquisition, M.G. All authors have read and agreed to the published version of the manuscript.

**Funding:** This work was funded by a grant from the CSAA Insurance Group.

**Institutional Review Board Statement:** Not Applicable.

**Informed Consent Statement:** Not Applicable.

**Data Availability Statement:** Due to the nature of this research, participants of this study did not agree for their data to be shared publicly, so supporting data are not available.

**Conflicts of Interest:** Y.J. and A.D. were employed by the funding company at the time research was conducted.

## Note

1    https://www.iii.org/fact-statistic/facts-statistics-auto-insurance#Auto%20claims (accessed on 20 May 2022).

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
