# Peer review of "Nightly Automobile Claims Prediction from Telematics-Derived Features: A Multilevel Approach"

_risks, doi:10.3390/risks10060118_

Round 1
Reviewer 1 Report
Please find in the attached file my observations.

Reviewer 2 Report
Thank you for allowing me to review this interesting paper and research. My proposals to improve the paper are:
- add 5 most recent citations (literature not older than 2021) in the 1.3 section
- move from 1.3 section all information that consists of methods and data; therefore, move them to section 2. For example, paragraphs from lines 74 to 91
- revise the whole section 1.3.2 while it is not clear enough to the reader. For example, the sentence between lines 105-110 is too long
- to check the language and prepositions, like an, etc. For example lines: 118, 278, 284, 317,...
- state all abbreviations in their whole meanings when it arises for the first time: XGBoost, CAN, SMOTE, IID, etc.
- use in the paper SI units (International System of Units), not the US ones. Use km/h instead of speed, km instead of miles (Tables 1, 2,...), etc.
Good luck.
Reviewer
Reviewer 3 Report
I recommend the article for publication after incorporating some comments. By its nature, the article belongs thematically to the journal Risks.
- A large part of the article is focused on the explanation of some statistical phenomena, characteristics and approaches. However, emphasis should be placed especially on the risk from the point of view of automobile insurance.
- Greater discussion should be devoted to the choice of input variables. Since the mean AUROC is between 0.59 and 0.7 (plus / minus the standard deviation), wouldn't it be interesting to change the input variables and see if the result doesn't work better?
- The final discussion indicates precisely the issues that I would expect to be discussed in the article.
- Graphic and typographic processing should be better. Sometimes the sentence starts with a lowercase letter, the pictures are not successful.
Round 2
Reviewer 1 Report
The Authors have replied to most of my points. It still remains the fact that results are based only on few claims thus the results are not supported by a well designed experiment. However, the paper is well written and presents some new ideas for future research.